# Genetically Higher Level of Mannose Has No Impact on Cardiometabolic Risk Factors: Insight from Mendelian Randomization

**DOI:** 10.3390/nu13082563

**Published:** 2021-07-27

**Authors:** Mohsen Mazidi, Abbas Dehghan, Maciej Banach

**Affiliations:** 1Medical Research Council Population Health Research Unit, University of Oxford, Oxford OX3 7LF, UK; 2Clinical Trial Service Unit and Epidemiological Studies Unit (CTSU), Nuffield Department of Population Health, University of Oxford, Oxford OX3 7LF, UK; 3Department of Twin Research & Genetic Epidemiology, King’s College London, South Wing St Thomas’, London SE1 7EH, UK; 4MRC-PHE Centre for Environment and Health, School of Public Health, Imperial College London, London W2 1PG, UK; a.dehghan@imperial.ac.uk; 5Department of Preventive Cardiology and Lipidology, Chair of Nephrology and Hypertension, Medical University of Lodz, 93-338 Lodz, Poland; 6Cardiovascular Research Centre, University of Zielona Gora, 65-046 Zielona Gora, Poland

**Keywords:** mendelian randomization, mannose, cardiovascular disease, cardio metabolic risk factors

## Abstract

**Background:** There is a handful of controversial data from observational studies on the serum levels of mannose and risks of coronary artery disease (CAD) and other cardiometabolic risk factors. We applied Mendelian Randomization (MR) analysis to obtain estimates of the causal effect of serum mannose on the risk of CAD and on cardiometabolic risk factors. **Methods**: Two-sample MR was implemented by using summary-level data from the largest genome-wide association studies (GWAS) conducted on serum mannose and CAD and cardiometabolic risk factors. The inverse variance weighted method (IVW) was used to estimate the effects, and a sensitivity analysis including the weighted median (WM)-based method, MR-Egger, MR-Pleiotropy RESidual Sum and Outlier (PRESSO) were applied. Radial MR Methods was applied to remove outliers subject to pleiotropic bias. We further conducted a leave-one-out analysis. **Results**: Mannose had no significant effect on CAD (IVW: odds ratio: 0.96 (95% Confidence Interval (95%CI): 0.71−1.30)), total cholesterol (TC) (IVW: 95%CI: 0.60−1.08), low density lipoprotein (LDL) (IVW: 95%CI = 0.68−1.15), high density lipoprotein (HDL) (IVW: 95%CI = 0.85−1.20), triglycerides (TG) (IVW: 95%CI = 0.38−1.08), waist circumference (WC) (IVW: 95%CI = 0.94−1.37), body mass index (BMI) (IVW: 95%CI = 0.93−1.29) and fasting blood glucose (FBG) (IVW: 95%CI = 0.92−1.33), with no heterogeneity for CAD, HDL, WC and BMI (all *p* > 0.092), while a significant heterogeneity was observed for TC (IVW: Q = 44.503), LDL (IVW: Q = 33.450), TG (IVW: Q = 159.645) and FBG (IVW: Q = 0. 32.132). An analysis of MR-PRESSO and radial plots did not highlight any outliers. The results of the leave-one-out method demonstrated that the links were not driven by a single instrument. **Conclusions:** We did not find any effect of mannose on adiposity, glucose, TC, LDL, TG and CAD.

## 1. Introduction

Mannose, a C2epimer of D-glucose, is considered a natural bioactive monosaccharide and one of the major carbohydrates of glycoprotein participating in N-glycosylation reactions [1]. Mannose can be obtained from both plants and microorganisms, while the majority of mannose comes from endogenous production [2,3]. Moreover, the physiological blood concentration of mannose in humans is about 50 microM derived from N-glycon processing [4]. Importantly, observational studies have shown that plasma mannose levels are linked to the risk of several chronic diseases including type 2 diabetes (T2D), cardiovascular disease (CVD) and albuminuria [5], while these studies still constitute only a handful and are controversial [6,7]. Mannose is naturally found in high amounts in many fruits such as apples, oranges and peaches, as well as blueberries and cranberries. Vegetables, including green beans, cabbage and broccoli, can also be a source of this bioactive.

A limited number of studies evaluated the association between mannose and cardiometabolic risk factors [6,7,8]. Mannose supplementation in animal studies showed a resistance to weight gain, liver steatosis and glucose intolerance caused by a high-fat diet. The results of a human epidemiological study showed that the body mass index (BMI) and glucose were positively correlated with plasma mannose levels [6,7]. The results of two cross-sectional studies demonstrated that plasma mannose levels were associated with an incident T2D risk [5,9]. While the authors were not able to rule out the chance of a residual bias, confounding factors and reverse causation, it has been reported that higher levels of mannose have been associated with higher glucose, HbA1c, a high waist-to-hip ratio, a higher risk of coronary heart disease (CHD), heart failure and mortality [5], with no association between mannose levels and low-density lipoproteins (LDL), cholesterol, BMI and insulin resistance [5]. It has also been reported that the plasma concentration of mannose was higher in participants with normal glucose tolerance than impaired glucose tolerance; a strong correlation between the mannose level and fasting blood glucose (FBG) was additionally reported [10]. In the same line, a positive correlation between the plasma mannose level and BMI and FBG was observed [11,12]. In a rodent model fed with a high-fat diet, mannose supplementation prevented weight gain and improved glucose hemostatis, while those animals had a significantly higher level of total cholesterol and high-density lipoprotein (HDL) [13].

Despite scattered evidence on the association between mannose levels and cardiometabolic health, the observational setting of the studies could not provide any evidence of the causality of the associations. In this study, we have implemented a Mendelian Randomization (MR) analysis to circumvent limitations of observational studies (residual bias, confounding factors and reverse causation) [14] and assess the potential causal effect of mannose by using genetic variants that are associated with an exposure (i.e., mannose) as instruments to test for associations with an outcome [14].

## 2. Methods

### 2.1. Study Design

A two-sample MR study design was used. Summary statistics were obtained from the largest genome wide association studies (GWAS) on mannose and interested outcomes. We applied methods to estimate the unbiased effect of mannose on the risk of coronary artery disease (CAD) and cardiometabolic risk factors.

### 2.2. Genetic Instruments for Mannose

After linkage disequilibrium (LD) clumping (0.0001) on all SNPs with a *p*-value under 1 × 10^−6^, we ended up with 19 independent SNPs, which were identified as being associated with a mannose concentration among 7824 adult samples of European ancestry (Table 1). More information can be found elsewhere [15]. F-statistic was used to assess the strength of the instruments and was calculated using the following equation: F = (*R*2/*k*)/([1 − *R*2]/[*n* − *k* − 1]), where *R2* is the proportion of the mannose levels accounted for by the SNP (SD = 0.127), *k* is the number of instruments used in the model and *n* is the sample size [16].

### 2.3. Association of Genetic Instruments with Outcome

The association of genetic instruments with CAD was retrieved from the largest GWAS on CAD including up to 76,014/264,785 cases/controls. The study had meta-analyzed results from a case (*n* = 60,801)-control (*n* = 123,504) GWAS on 1000 Genomes data from CARDIoGRAMplusC4D Consortium, the UK Biobank SOFT CAD study (cases *n* = 10,801, controls *n* = 137,371), and two small case (*n* = 4120)-control (*n* = 3910) studies from Germany and Greece [17]. Participants for the CARDIoGRAMplusC4D study were mainly of European descent (77%), and the case definition for CAD, or MI, was based on medical records, clinical diagnosis or any procedures that indicated CAD [17,18].

The association of genetic instruments with lipids was obtained from GWAS conducted by the Global Lipid Genetics Consortium (GLGC) (up to 188,577 persons). GLGP has conducted GWAS on LDL, TG, HDL and TC [18,19] on 1000 Genome imputed data GLGC meta-analyzed GWAS from 47 studies that were genotyped using genome-wide association study arrays (*n* = 94,595) or the Metabochip array (*n* = 93,982). The Metabochip was designed to facilitate a cost-effective follow-up of nominal associations for metabolic diseases and cardiovascular traits and to enhance the fine mapping of established loci. The goal of the 1000 Genomes Project was to find common genetic variants with frequencies of at least 1% in the populations studied. The 1000 Genomes Project took advantage of developments in sequencing technology, which sharply reduced the cost of sequencing. It was the first project to sequence the genomes of a large number of people in order to provide a comprehensive resource on human genetic variation [18,19]. In most included studies, blood lipid levels were measured after >8 h of fasting, and those who were on lipid-lowering medications were excluded. GWAS included a linear regression analysis adjusted for age, age-squared, sex and principal components. Lipid levels were inverse normal-transformed within each cohort, and the results were combined using a fixed effect meta-analysis [19].

For fasting glucose, we used GWAS published by the MAGIC consortium on fasting glucose markers. All included studies (*n* = 21) were of European descent [20]. Information on the BMI and waist circumference (WC) was obtained from the Genetic Investigation of Anthropometric Traits (GIANT) Consortium, using GWAS data on 339,224 individuals [21].

### 2.4. Mendelian Randomization Analysis

We conducted a Mendelian randomization study using 19 genetic instruments. The causal estimates for all instruments were combined using the inverse variance weighted method as implemented in the TwoSampleMR package running under R. Heterogeneity was assessed using the Q value for IVW. To rule out a pleiotropic effect of the instruments (pleiotropy refers to the phenomenon of a gene or genetic variant affecting more than one phenotypic trait) on the final effect estimate, we conducted a sensitivity analysis including a weighted median and MR-egger. Moreover, we conducted a leave-one-out analysis (leave-one-out performs by excluding one variant at each analysis; it is useful for investigating the influence of each variant on the overall effect-size estimate and for identifying influential variants) to identify instruments that might drive the MR results. Bootstrapping was applied to estimate the confidence intervals (CIs) for inverse variance estimates [22]. The weighted median (WM) estimate was used as the first sensitivity analysis since it provides correct estimates as long as SNPs accounting for ≥50% of the weight are valid instruments. MR-egger, which was used as the second sensitivity analysis, has an ability to provide correct estimates even when all SNPs are invalid instruments as long as the assumption of instrument strength independent of direct effect (InSIDE) is satisfied [22]. The average directional pleiotropy across genetic variants was assessed from the *p*-value of the intercept term from MR-Egger [22].

To assess the heterogeneity between individual genetic variant estimates, we used the Q′ heterogeneity statistic [23] and the MR pleiotropy residual sum and outlier (MR-PRESSO) test [23]. The MR-PRESSO framework applies residual diagnostics to detect genetic instruments that are outliers with respect to the model fit. This is done by fitting a weighted regression model and estimating the residuals. A global heterogeneity index that represents all genetic instruments is tested to find out whether there is in general more heterogeneity than expected by chance in all the data. A local test could later be applied to focus on individual instruments and detect if any of them are outliers [24]. Further, we applied the MR-Robust Adjusted Profile Score (RAPS), which corrects for pleiotropy using robust adjusted profile scores. We performed radial variants of the IVW and MR-Egger models for a better illustration of the IVW and MR-Egger results. For radial IVW, an intercept is omitted in contrast to radial MR-Egger, as in the conventional setting. The advantage is that the genetic instruments that are subject to the pleiotropic effect are highlighted as potential outliers, based on residual values. This is proportional to the contribution of each instrument to the global heterogeneity. All analyses were done using R, version 3.4.2 R Core Team (2017) [25]. In this study, we examined the effect of serum mannose on seven traits (CAD, total cholesterol [TC], triglyceride [TG], LDL, HDL, WC and BMI); therefore, the significance threshold was adjusted using the Bonferroni correction (α = 0.05/7 = 0.0071).

In an alternative approach for addressing the potential pleiotropic effect of the genetic instruments, we repeated the IVW analysis, excluding SNPs with potentially pleiotropic effects, and we compared the results with the original analysis. We used Ensembl release (http://useast.ensembl.org/index.html) (accessed on 6 December 2019), which includes a base of SNP phenotypes, to assess the exclusion-restriction analysis.

### 2.5. Ethics

The analysis conducted in this project merely benefits from published or publicly available summary data, which means that we did not have to contact any of the participants in the study or collect any original data. Ethical approval for the participating studies, including the information on their informed consent, could be found in the original publications. The study follows the ethical guidelines according to the 1975 Declaration of Helsinki.

## 3. Results

The list of all genetic instruments for mannose is shown in Table 1. All instruments had F statistics of more than 10, ranging from 19 to 79 [16]. The results of the MR analysis, expressed as the *beta*-coefficient for 1 standard deviation (SD) increase in mannose levels, are presented in Table 2. Mannose had no significant effect on CAD (IVW: odds ratio: 0.96, 95% confidence interval [95%CI] = 0.71–1.30; *p* = 0.835), TC (IVW: 95%CI = 0.60–1.08; *p* = 0.161), LDL (IVW: 95%CI = 0.68–1.15; *p* = 0.375), HDL (IVW: 95%CI = 0.85–1.20; *p* = 0.896), TG (IVW: 95%CI = 0.38–1.08; *p* = 0.097), WC (IVW: 95%CI = 0.94–1.37; *p* = 0.162), BMI (IVW: 95%CI = 0.93–1.29; *p* = 0.234), FBG (IVW: 95%CI = 0.92–1.33; *p* = 0.259). IWV estimates showed no heterogeneity for CAD, HDL, WC and BMI (all *p* > 0.092, Table 1). Additionally, a radial analysis conducted to aid in the detection of outlying variants (Figure 1)—IVW Radial MR and Egger Radial MR—delineated no instrumental variables as potential outliers. A significant heterogeneity was observed for TC (IVW: Q = 44.503), LDL (IVW: Q = 33.450), TG (IVW: Q = 159.645) and FBG (IVW: Q = 32.132).

The analysis of MR-PRESSO has not highlighted any outlier instruments for any of the analyses. The Egger intercept was only significant for BMI; however, Egger estimates did not show any significant causal effect. The results of the leave-one-out method demonstrated that the links were not driven by the single SNPs.

## 4. Discussion

We have conducted an MR analysis to evaluate the causal effect of mannose metabolites on cardiovascular and cardiometabolic risk factors. Our results showed that subjects with a genetically higher level of mannose had no significant association with levels of adiposity (BMI, WC) and FBG. We also did not find any causal effect of mannose on HDL- and LDL-cholesterol, TG and TC. There was no evidence to imply that pleiotropy, heterogeneity or outliers had biased results.

Mannose, as a critical hexose for glycoprotein synthesis, is considered one of the metabolites in the liver of obese subjects [8]. Diet is not the main source of circulating mannose since diet cannot ensure the physiological needs of the body for glycosylation. Endogenous production in the liver is the main source of mannose production in the human body [10]. Plasma mannose levels are decreased by the intravenous administration of insulin, which stimulates glycogen synthesis [26]; however, ingested glucose leads to elevated plasma mannose levels via increasing glucokinase [27].

Observational studies have reported an association between mannose levels and a number of cardiometabolic traits and disorders. However, we did not find any causal relation between mannose levels and cardiometabolic traits and disorders. Observational studies are known to be subject to well-characterized biases such as confounding and reverse causation. It is likely that the reported associations are biased and are not purported to be causal. For instance, animal and human studies have shown that mannose levels are associated with glucose levels [27,28]. A recent report on Japanese individuals revealed that plasma mannose levels were significantly correlated with BMI (R = 0.253, *p*-value = 0.0283), fasting plasma glucose (R = 0.371, *p*-value = 0.0010) and fasting insulin (R = 0.295, *p*-value = 0.0102) [11]. Several other limitations might explain the controversy. First, insulin sensitivity as an important determinant of the fasting mannose level was not measured by the hyperinsulinemic-euglycemic clamp. Second, the relationship between glycogenolysis and endogenous glucose production is controversial since the measurement of glycogenolysis was not conducted by methods using glucose isomers. Importantly, the cross-sectional design of the study cannot evaluate the causal association.

Another study with the aim of evaluating physiological changes in the circulating mannose level in 273 normal, glucose-intolerant and diabetic patients demonstrated that the fasting plasma mannose (FPM) levels of diabetic subjects were higher than those of normal subjects, and there was a strong positive correlation between FPM and fasting plasma glucose levels (r = 0.745) [10]. It should be noted that the enzymatic methods to evaluate plasma and serum mannose levels in the study were not completely accurate. The effects of concomitant drug therapy, dietary change, exercise and aging should be considered. Recently, a cell-specific integrated network analysis reported that mannose was significantly correlated with BMI (r = 0.34, *p*-value < 0.05) and insulin secretion (r = −0.13, *p*-value < 0.05). Plasma mannose levels were also increased in response to obesity, and a strong correlation was found between glucose and mannose (r = 0.64, *p*-value < 0.05) [8].

It is reported that mannose production can be influenced by glucose-6-phosphate (G6P) metabolism [29,30,31]. The absent activity of G6Pase leads to a high content of G6P and therefore increased mannose levels. Due to the fact that the accumulation of G6P is expected to be directed toward lactate production and acetyl-CoA, which are substrates for free fatty acids and triglycerides, the mannose level can be closely linked with lactate or triglycerides [32]. It is widely known that increased fasting or nonfasting levels of triglycerides contribute to an elevated CVD risk [33]. Triglycerides-rich remnants cause an increase in endothelial dysfunction, which is considered as the first step in atherogenesis. Turnover of lipoprotein may be echoed by the mannose concentration. The oligosaccharides attached to the lipoprotein include both mannose-rich and complex carbohydrate chains. It may be hypothesized that increased very-low-density lipoprotein (VLDL) and apolipoprotein B (apoB) secretions depend on an increased mannose utilization and availability of mannose-6 phosphate [28]. In three cohort studies, it was reported that plasma mannose levels were significantly associated with metabolic syndrome (MetS) factors [4]. However, in accordance with our results, it was not significantly correlated with LDL-C levels. Furthermore, these studies evaluated the genetic risk score for T2D, BMI and the waist-to-hip ratio, showing no association with BMI and insulin resistance [5]. Moreover, a mannose supplementation study has shown that the levels of triglycerides, HDL-cholesterol and LDL-cholesterol were not statistically different before and after one week of mannose intake (0.1 g/kg BW twice daily) [34]. An animal study on a C57BL6 mouse model of diet-induced obesity showed that mannose supplementation of high-fat-diet (HFD)-fed mice prevented weight gain and lowered adiposity [12]. To be precise, the rates of triglyceride synthesis and VLDL clearance were not changed by HFD+mannose compared to HFD. Moreover, LDL (low-density lipoprotein) and VLDL (very-low-density lipoprotein) were not altered between the two groups, which was compatible with our results [13]. However, as opposed to our findings, a relatively small study on 20 diabetic patients revealed that serum mannose was positively correlated with blood glucose (r = 0.758, *p*-value = 0.012), triglycerides (r = 0.478, *p*-value = 0.023) and inversely with HDL-cholesterol (r = −0.427, *p*-value = 0.022) [28], although this study might suffer from a small sample size and power and a chance of residual bias.

Our analysis has certain strengths and limitations. Our approach, MR, is a powerful tool for assessing causality, which makes our study superior to all former observational studies. Moreover, we benefited from the largest GWAS on mannose and outcomes. However, MR analysis is known to lack statistical power. To address this, we used the largest genetic association studies available. We would also like to highlight a chance of horizontal pleiotropy in our analysis.

In conclusion, we did not find any effect of the circulating level of the plasma mannose on adiposity, glucose, and LDL, TG and TC. The associations that were reported in previous studies are either confounded, and the true effect estimates are either null or are smaller than what is now reported.

## Figures and Tables

**Figure 1 nutrients-13-02563-f001:**
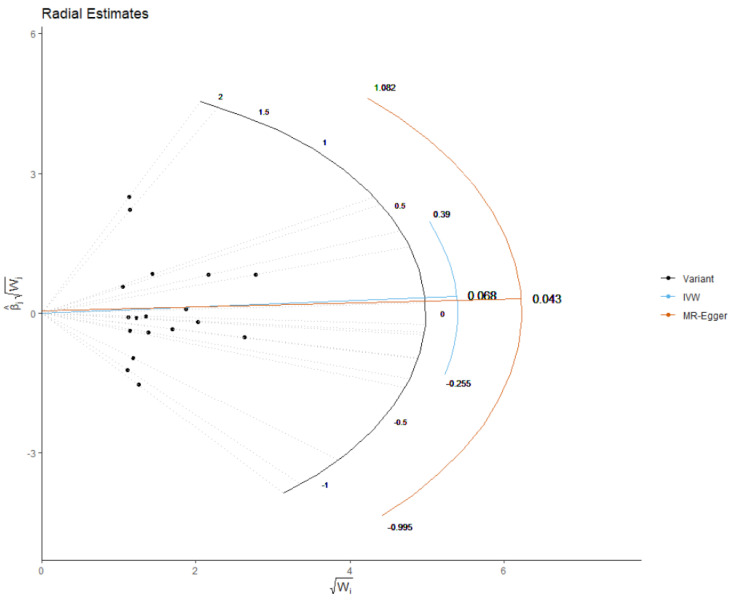
Radial plots to visualize individual outlier single nucleotide polymorphisms (SNPs) in the Mendelian randomization (MR) estimates for effect of the plasma mannose and risk of CHD. The radial curve displays the ratio estimate for each genetic variant, as well as the overall inverse-variance weighted (in blue) and MR Egger estimate (in orange). Black dots show valid SNPs. IVM: inverse variance weighted method.

**Table 1 nutrients-13-02563-t001:** Summary results of the genetic loci of serum mannose.

SNP	GX	GX SE	EA	OA	EAF
rs1275504	0.0168	0.0025	A	T	0.2976
rs11676911	−0.0429	0.0084	A	G	0.0216
rs4980268	0.019	0.0039	A	G	0.9184
rs12474099	−0.0202	0.0043	A	G	0.0649
rs7747345	0.0122	0.0026	A	T	0.8153
rs13116473	−0.0448	0.0096	A	G	0.9853
rs10736	−0.0118	0.0026	T	C	0.2011
rs12414366	0.0214	0.0047	C	G	0.7696
rs16932803	−0.0121	0.0026	T	G	0.1673
rs1181723	0.0116	0.0026	A	G	0.2086
rs1841535	0.0252	0.0056	T	G	0.1482
rs991670	−0.0111	0.0025	T	C	0.6624
rs2272172	−0.0648	0.0145	T	C	0.0226
rs2270451	0.0152	0.0034	A	G	0.8941
rs7007251	−0.015	0.0034	A	G	0.1057
rs17061914	−0.0146	0.0033	A	G	0.8581
rs10982244	−0.0177	0.004	A	G	0.6098
rs2723713	−0.0356	0.008	T	C	0.0254

EA: effect allele; OA: other allele; EAF: effect allele frequency; GX: the per-allele effect on standard deviation units of the mannose; GX SE: standard error of GX; SNP: single nucleotide polymorphism.

**Table 2 nutrients-13-02563-t002:** Results of the MR analysis for all outcomes.

Exposures	MR				Heterogeneity			Pleiotropy		
	Method	Beta	SE	*p*-Value	Method	Q	*p*-Value	Intercept	SE	*p*-Value
**CAD**	**MR Egger**	−0.609	0.346	0.097	**MR Egger**	15.683	0.475	0.0122	0.006	0.084
	**WM**	−0.189	0.199	0.342						
	**IVW**	−0.031	0.153	0.835	**IVW**	19.060	0.324			
	**RAPS**	−0.095	0.153	0.532						
**TC**	**MR Egger**	−0.569	0.409	0.182	**MR Egger**	42.170	0.0003	0.006	0.007	0.360
	**WM**	−0.010	0.155	0.946						
	**IVW**	−0.212	0.151	0.161	**IVW**	44.503	0.0002			
	**RAPS**	−0.242	0.157	0.122						
**LDL**	**MR Egger**	−0.709	0.336	0.051	**MR Egger**	27.361	0.037	0.010	0.005	0.077
	**WM**	−0.044	0.140	0.753						
	**IVW**	−0.119	0.134	0.375	**IVW**	33.450	0.009			
	**RAPS**	−0.140	0.131	0.283						
**HDL**	**MR Egger**	0.308	0.235	0.207	**MR Egger**	13.746	0.617	−0.005	0.004	0.191
	**WM**	0.068	0.127	0.589						
	**IVW**	0.011	0.088	0.896	**IVW**	15.605	0.551			
	**RAPS**	0.007	0.097	0.939						
**TG**	**MR Egger**	−0.522	0.729	0.484	**MR Egger**	159.485	<0.001	0.001	0.012	0.900
	**WM**	−0.13	0.137	0.337						
	**IVW**	−0.436	0.263	0.097	**IVW**	159.645	<0.001			
	**RAPS**	−0.143	0.160	0.370						
**WC**	**MR Egger**	−0.094	0.257	0.717	**MR Egger**	25.005	0.094	0.004	0.004	0.353
	**WM**	0.150	0.116	0.198						
	**IVW**	0.133	0.095	0.162	**IVW**	26.345	0.092			
	**RAPS**	0.110	0.099	0.267						
**BMI**	**MR Egger**	−0.318	0.199	0.128	**MR Egger**	20.020	0.273	0.007	0.003	0.037
	**WM**	0.086	0.101	0.391						
	**IVW**	0.098	0.083	0.234	**IVW**	26.004	0.099			
	**RAPS**	0.089	0.080	0.269						
**FBG**	**MR Egger**	−0.177	0.236	0.464	**MR Egger**	29.217	0.032	0.005	0.004	0.210
	**WM**	−0.018	0.103	0.855						
	**IVW**	0.106	0.094	0.259	**IVW**	32.132	0.021			
	**RAPS**	0.046	0.095	0.627						

WM: Weighted median, IVW: Inverse variance weighted, SE: standard error, beta: beta-coefficients, CAD: coronary artery disease, TC: total cholesterol, LDL: low-density lipoproteins, HDL: high-density lipoprotein, MR: mendelian randomization, TG: triglyceride, WC: waist circumference, BMI: body mass index, FBG: fasting blood glucose, RAPS: Robust Adjusted Profile Score: Q: Cochran’s Q.

## Data Availability

The data underlying this article will be shared on reasonable request to the corresponding author.

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
