# Peer review of "Genetically Higher Level of Mannose Has No Impact on Cardiometabolic Risk Factors: Insight from Mendelian Randomization"

_nutrients, 2021, doi:10.3390/nu13082563_

Round 1

Reviewer 1 Report

This study addresses the relatively unstudied effects of mannose on CVD and CAD.  The methods appear to be sound and the topic is of considerable interest.  The methods are not, however, standard and the use of the model should be expanded to better explain the statistical underpinnings of the model.

The other issue is that there are numerous errors in English that make the manuscript difficult to read.  For example, the first sentence of the abstract is poor English and hard to read.  Attention needs to be paid throughout the manuscript to simply staying in the same tense, having singular and plurals agree and proper articles are essential.

Abstract: Background: There is a handful of  controversial data from observational             23

studies on serum levels of mannose and risks of coronary artery disease (CAD) and other

cardiometabolic risk factors.                                                                                                  24

while  the majority of mannose comes from endogenous-51

or on the Metabochip array (n=93,982) with imputation -121

An explanation of what Metabochip refers to is needed

 to the 1000 Genomes Project reference were studied

A reference and explanation of this Genomes project is needed

Better explanation of the statistics are needed here:

We assessed the heterogeneity using Q value for IVW. To address potential 140 effect of pleiotropic variants on the final effect estimate, we conducted sensi-141 tivity analysis including weighted median and MR-egger. Moreover, we con-142 ducted leave-one-out analysis to identify instruments that might drive the MR 143 results. Bootstrapping was applied to estimate the confidence intervals (CIs) 144 for inverse variance estimates (22) The weighted median (WM) estimate was 145 used as the first sensitivity analysis since it provides correct estimates as long 146 as SNPs accounting for ≥50% of the weight are valid instruments. MR-egger 147 that was used as the second sensitivity analysis has an ability to provide cor-148 rect estimates even when all SNPs are invalid instruments as long as the as-149 sumption of instrument strength independent of direct effect (InSIDE) is sat-150 isfied (22). Average directional pleiotropy across genetic variants was as-151 sessed from the p-value of the intercept term from MR-Egger (22). 15

Line 241-242 For instance,

XXX et al have shown that mannose levels are associated with glucose 242 levels (27,28).

XXX needs to be changed

Author Response

This study addresses the relatively unstudied effects of mannose on CVD and CAD.  The methods appear to be sound, and the topic is of considerable interest.  The methods are not, however, standard and the use of the model should be expanded to better explain the statistical underpinnings of the model.

Thank you! We have provided further info and explanation for our methods. It is worth mentioning that each section of the method comes with the references therefore readers are guided to a proper reference in case of interest. What is more, we were asked to decrease the overlapping ratio in the methods, therefore we do not want to repeat the parts of the methodology that were described in detail previously.

The other issue is that there are numerous errors in English that make the manuscript difficult to read.  For example, the first sentence of the abstract is poor English and hard to read.  Attention needs to be paid throughout the manuscript to simply staying in the same tense, having singular and plurals agree and proper articles are essential.

Abstract: Background: There is a handful of controversial data from observational       23

studies on serum levels of mannose and risks of coronary artery disease (CAD) and other

cardiometabolic risk factors.                                                                                                   24

while  the majority of mannose comes from endogenous-51

Thanks for the comment, English grammar, spelling, and punctuation has been revised all over the manuscript.

or on the Metabochip array (n=93,982) with imputation -121

An explanation of what Metabochip refers to is needed

Thanks for the comment, now we have provided further information on the Metabochip:

Metabochip is designed to facilitate cost-effective follow-up of nominal associations for metabolic diseases andcardiovascular traits and to enhance fine mapping of established loci.”

to the 1000 Genomes Project reference were studied

A reference and explanation of this Genomes project is needed

Thanks for the comment, now we have provided further information on the 1000 Genomes Project:

The goal of the 1000 Genomes Project was to find common genetic variants with frequencies of at least 1% in the populations studied. The 1000 Genomes Project took advantage of developments in sequencing technology, which sharply reduced the cost of sequencing. It was the first project to sequence the genomes of a large number of people, to provide a comprehensive resource on human genetic variation.”

This information was also suitably referenced.

Better explanation of the statistics are needed here:

We assessed the heterogeneity using Q value for IVW. To address potential 140 effect of pleiotropic variants on the final effect estimate, we conducted sensi-141 tivity analysis including weighted median and MR-egger. Moreover, we con-142 ducted leave-one-out analysis to identify instruments that might drive the MR 143 results. Bootstrapping was applied to estimate the confidence intervals (CIs) 144 for inverse variance estimates (22) The weighted median (WM) estimate was 145 used as the first sensitivity analysis since it provides correct estimates as long 146 as SNPs accounting for ≥50% of the weight are valid instruments. MR-egger 147 that was used as the second sensitivity analysis has an ability to provide cor-148 rect estimates even when all SNPs are invalid instruments as long as the as-149 sumption of instrument strength independent of direct effect (InSIDE) is sat-150 isfied (22). Average directional pleiotropy across genetic variants was as-151 sessed from the p-value of the intercept term from MR-Egger (22). 15

Thank you. We have updated it accordingly:

We assessed the heterogeneity using Q value for IVW. To address potential effect of pleiotropic variants (pleiotropy refers to the phenomenon of a gene or genetic variant affecting more than one phenotypic trait) on the final effect estimate, we conducted sensitivity analysis including weighted median and MR-egger. Moreover, we conducted leave-one-out analysis (leave-one-out performs by excluding one variant at each analysis. It is useful to investigate the influence of each variant on the overall effect-size estimate and to identify influential variants) to identify instruments that might drive the MR results. Bootstrapping was applied to estimate the confidence intervals (CIs) for inverse variance estimates (22) The weighted median (WM) estimate was used as the first sensitivity analysis since it provides correct estimates as long as SNPs accounting for ≥50% of the weight are valid instruments. MR-egger that was used as the second sensitivity analysis has an ability to provide correct estimates even when all SNPs are invalid instruments as long as the assumption of instrument strength independent of direct effect (InSIDE) is satisfied (22). Average directional pleiotropy across genetic variants was assessed from the p-value of the intercept term from MR-Egger (22).

Line 241-242 For instance,

XXX et al have shown that mannose levels are associated with glucose 242 levels (27,28).

XXX needs to be changed

Thank you, it has been revised accordingly.

  1. Line 54 - it should be rather 50 microM (M not m)

  1. Introduction - please extend the Introduction providing data on dietary mannose. What are the main dietary sources of this sugar? Could you please find some data on the correlation between dietary intake of mannose and the risk of cardiometabolic disorders.

  1. Line 91-CAD?

  1. Table 2-what is CAD? I suggest to add a separate paragraph with all abbreviations used in the manuscript. 

  1. Line 242-what is XXX?

1.     Thank you. It has been revised.

2.     Dietary sources of the mannose have been added to the manuscript, further we have mentioned the effect of the mannose supplementation in mice.

3.     Thank you. The abbreviation has been explained.

4.     Now we have defined all the abbreviations in the table.

5.     Thank you. It has been revised.

Reviewer 2 Report

  1. Line 54 - it should be rather 50 microM (M not m)
  2. Introduction - please extend the Introduction providing data on dietary mannose. What are the main dietary sources of this sugar? Could you please find some data on the correlation between dietary intake of mannose and the risk of cardiometabolic disorders.
  3. Line 91-CAD?
  4. Table 2-what is CAD? I suggest to add a separate paragraph with all abbreviations used in the manuscript. 
  5. Line 242-what is XXX?

Author Response

  1. Line 54 - it should be rather 50 microM (M not m)

  1. Introduction - please extend the Introduction providing data on dietary mannose. What are the main dietary sources of this sugar? Could you please find some data on the correlation between dietary intake of mannose and the risk of cardiometabolic disorders.

  1. Line 91-CAD?

  1. Table 2-what is CAD? I suggest to add a separate paragraph with all abbreviations used in the manuscript. 

  1. Line 242-what is XXX?

1.     Thank you. It has been revised.

2.     Dietary sources of the mannose have been added to the manuscript, further we have mentioned the effect of the mannose supplementation in mice.

3.     Thank you. The abbreviation has been explained.

4.     Now we have defined all the abbreviations in the table.

5.     Thank you. It has been revised.
